# A Comparison of EQ-5D-3L, EQ-5D-5L, and SF-6D Utilities of Patients with Musculoskeletal Disorders of Different Severity: A Health-Related Quality of Life Approach

**DOI:** 10.3390/jcm11144097

**Published:** 2022-07-15

**Authors:** Nikolaos Kontodimopoulos, Eleni Stamatopoulou, Sousana Gazi, Dimitra Moschou, Michail Krikelis, Michael A. Talias

**Affiliations:** 1Department of Health Economics, Medical School, National and Kapodistrian University of Athens, 11527 Athens, Greece; 2Department of Biomedical Sciences, University of West Attica, 12243 Athens, Greece; elenastamatopoyloy@gmail.com; 3Rheumatology Department, General Hospital of Attica “KAT”, 14561 Athens, Greece; susanagazi@yahoo.com (S.G.); dimitramoschou@hotmail.com (D.M.); michael.krikelis@gmail.com (M.K.); 4Healthcare Management Postgraduate Program, School of Economics & Management, Open University of Cyprus, Nicosia 2220, Cyprus; michael.talias@ouc.ac.cy

**Keywords:** EQ-5D-5L, EQ-5D-3L, SF-6D, utilities, health-related quality of life, musculoskeletal disorders, rheumatoid arthritis, psoriatic arthritis, ankylosing spondylitis, osteopenia–osteoporosis

## Abstract

This study compares EQ-5D-3L, EQ-5D-5L, and SF-6D utilities in patients with different musculoskeletal (MSK) disorders, also differing in disease severity as defined by valid clinical indexes. Utilities were measured from a cross-sectional sample of rheumatoid arthritis (N = 114), psoriatic arthritis (N = 57), ankylosing spondylitis (N = 49), and osteopenia/osteoporosis (N = 95) patients. For the first three groups, disease activity (severity) was measured with the DAS-28, DAPSA, and BASDAI clinical indexes, respectively. Mean differences and effect sizes were measured, and agreement between utilities was estimated with the intraclass correlation coefficient and Bland–Altman plots. Higher agreement was observed between EQ-5D-5L and SF-6D, compared to EQ-5D-3L and SF-6D, in all MSK disorder groups and severity levels. In groups with moderate to high severity, agreement between EQ-5D-3L/SF-6D and EQ-5D-5L/SF-6D was between low and fair, and both EQ-5D-3L and 5L utilities were lower than SF-6D (*p* < 0.001). On the other hand, in remission or low activity groups, agreement was excellent, and SF-6D utilities were again typically higher than EQ-5D-3L/5L, but not significantly. In more severe patients, SF-6D generated significantly higher utilities than EQ-5D-3L and 5L, which is consistent with most previous studies. Such discrepancies could have implications on economic evaluations of interventions targeting patients with MSK disorders.

## 1. Introduction

Global population growth and increased life expectancy have made people increasingly susceptible to non-communicable diseases, such as musculoskeletal (MSK) disorders. This generic term covers over 150 conditions ranging from short-term problems such as strains and sprains, to long-term or lifelong conditions, most of which are associated with intense and persistent pain and limitations in mobility, functioning, and work ability. MSK disorders have increased over the years and currently affect approximately 1.71 billion people [1], and account for more than 21% of worldwide disability—second only to mental and behavioral problems—with a significant cost to individuals and society through the associated healthcare needs [2].

A common MSK disorder is rheumatoid arthritis (RA), which is a chronic, inflammatory, autoimmune disorder primarily affecting joints, and carrying substantial physical and psychological morbidities, with a profound impact on health-related quality of life (HRQoL) [3]. Psoriatic arthritis (PsA), which is another chronic disorder, occurs in about 40% of people with psoriasis, and is often linked to other comorbidities, functional limitations, and psychosocial disability [4]. Ankylosing spondylitis (AS) is arthritis characterized by long-term inflammation of the spine’s joints, resulting in back pain, and occasionally, other joints such as the shoulders or hips are also involved [5]. Osteoporosis (OP) is a systemic skeletal disorder causing bone fragility and an increased risk of hip, vertebral, and wrist fractures. It is estimated that by 2040, more than 300 million people will be at high-fracture-risk due to osteoporosis [6]. Midway to OP is osteopenia (OsP), in which bone density is lower than normal, but not as severe, and treating it may slow the progression of bone loss leading to osteoporosis.

In the present fragile economic environment, studies comparing costs and outcomes between contending therapies should be integrated into clinical practice. Agencies such as the National Institute for Health and Care Excellence (NICE) in the UK recommend that preference-based utility measures be used in technology appraisals. Among the most widely used utility instruments for calculating QALYs are EQ-5D and SF-6D [7]. These two instruments, which differ in their descriptive systems and valuations attached to health states [8], have been compared extensively. Most studies show EQ-5D to be less sensitive in capturing health gains in less severe patients, due to many subjects scoring unity (i.e., perfect health) at baseline, thereby decreasing the likelihood that the instrument has accurately measured the intended dimension. This is known as the “ceiling effect”, and can create difficulties in accurately measuring central tendency and dispersion, as well as ranking individuals according to score. However, EQ-5D is better at detecting health gains in more severe states, as its lowest utility score is −0.59, compared to +0.30 for SF-6D. Hence, higher-severity individuals usually have significantly higher mean utilities from SF-6D, whereas lower-severity individuals from EQ-5D, a finding which has been confirmed in studies with numerous disease groups [9,10,11,12,13,14], as well as in general population samples [15,16,17,18].

The 5-level EQ-5D (EQ-5D-5L) was introduced in 2009 to improve discriminant validity and sensitivity to change, and to reduce ceiling effects compared to the original three-level version. To assess this, many studies have compared the measurement properties of the two EQ-5D versions, with the results generally confirming the expected outcomes, i.e., reduced ceiling effects, better discriminatory power, and increased convergent and known-groups validity for EQ-5D-5L [19,20,21,22,23,24,25]. Comparisons of EQ-5D-5L with SF-6D are also increasing in the literature [26,27,28,29], with most of the previous studies having made utility comparisons over entire samples, or by socio-demographic variables, rather than on the basis of valid external instruments or clinical indexes capable of capturing measurable differences. Hence, it is possible that utility discrepancies, which might exist within the portions of the samples differing in severity (therefore, also in health status), are often overlooked. In light of these arguments, the purpose of this study was to compare EQ-5D-3L, EQ-5D-5L, and SF-6D utilities in groups of patients with different musculoskeletal disorders, who also differ in their disease severity, as defined by clinical indexes used to determine activity levels.

## 2. Materials and Methods

### 2.1. Instruments

The EQ-5D-3L is a five-dimensional preference-based instrument covering mobility, self-care, usual activities, pain/discomfort, and anxiety/depression, and each dimension is measured on three response levels corresponding to no, some, or severe problems [30]. Following the development of a UK preference-based valuation based on time trade-off (TTO) utilities [31], it became the most widely used generic multi-attribute utility instrument, and eventually, the preferred instrument for health technology assessments in many countries. It has a measurement range of −0.59 to 1, as some health states are considered as worse than dead. The construct validity of the EQ-5D-3L, using the UK valuation, has been demonstrated in a large sample of the Greek general population [32]. The newer EQ-5D-5L instrument consists of the same five-dimension descriptive system, but each dimension has five, instead of three, levels, corresponding to no, slight, moderate, severe, and extreme problems, which help increase sensitivity and reduce ceiling effects. The Greek EQ-5D-5L, which is one of the 130 language versions available, was translated according to EuroQol guidelines, and its validity was demonstrated in the Greek general population [33]. At present, there is no valuation for either EQ-5D-3L or EQ-5D-5L derived from the Greek population; therefore, in the present study, the former was scored using the widely used algorithm derived from health state weights reported by Dolan [31], whereas the latter from the UK “crosswalk” value sets [34].

The SF-6D is a preference-based utility index which originates from the conversion of 11 items of the widely used Short Form 36 Health Survey into a health status classification system with six domains corresponding to physical functioning, role limitations, social functioning, pain, mental health, and vitality, and with four to six severity levels in each. It generates 18,000 unique health states, and the first scoring model was based on standard gamble (SG) utilities from a UK general population sample. Regression models were used to predict utility scores ranging from 0.30 to 1 for each possible health state [35]. The reliability and validity of the Greek SF-36 (ver. 1) were established in a large general population sample living in the greater Athens area [36]. The SF-6D, with the UK valuation, has been used to estimate utility scores in many Greek Studies [12,14,18,37].

The Disease Activity Score 28 joints (DAS-28) is used to determine RA disease severity. The variables for its calculation are the number of swollen and tender joints, the erythrocyte sedimentation rate (ESR), and the patients’ general health measured on a visual analogue scale (VAS) [38]. The scores range from 0.49 to 9.07, and a value >5.1 corresponds to high disease activity, between 3.2–5.1 to moderate activity, between 2.6–3.2 to low activity, and <2.6 to remission [39]. The Disease Activity Index for Psoriatic Arthritis (DAPSA) is calculated by the summation of five variables: tender and swollen joints, self-assessed disease activity, pain measured on a VAS, and C-reactive protein (CRP). Based on observational and survey data, and on clinical trials, the cut points for classification of PsA activity are: ≤4 (remission), 4–14 (low activity), 15–18 (moderate activity), and >28 (high activity) [40]. The Bath Ankylosing Spondylitis Disease Activity Index (BASDAI) is used to determine AS treatment effectiveness. It consists of a 0–10 scale measuring discomfort, pain, and fatigue in response to six questions pertaining to the major symptoms of AS: fatigue, spinal pain, joint pain or swelling, localized tenderness level, and duration of morning stiffness. The resulting 0–50 score from the first five questions is divided by 5, with scores of >4 suggesting suboptimal control of the disease [41].

### 2.2. Sample and Data Collection

A cross-sectional sample of consecutive patients clinically diagnosed with RA, PsA, AS, or OsP/OP was recruited from the Rheumatologic Outpatient Department of “KAT” General Hospital of Attica between March 2020 and September 2020. Patients not understanding Greek, less than 18 years old, with psychological or other disorders which might prevent survey completion, or suffering from severe comorbidities which could “override” the MSK disorder were excluded. Eligible patients agreeing to participate were interviewed face-to-face by a trained interviewer. The survey consisted of the EQ-5D-5L, SF-36 (ver. 1), and EQ-5D-3L instruments, completed in that order by all the patients. Disease activity was measured with DAS-28 for RA, DAPSA for PsA, and BASDAI for AS. Osteoporosis and fracture risk were established by a Bone Mineral Density (BMD) test, and T-scores (i.e., standardized scores with the mean equal to 50 and the standard deviation equal to 10) were used for comparisons with healthy young adults. Demographic and disease-related data, including gender, age, educational background, disease duration, BMI, and comorbidities (e.g., hypertension, diabetes, osteoarthritis), were recorded or extracted from patients’ medical files.

### 2.3. Ethical Issues

The study was approved by the Institutional Review Board (IRB)/Ethics Committee (EC) of “KAT” General Hospital of Attica (IRB/EC approval reference number: 227/10-6-2019). The research was carried out in accordance with the Declaration of Helsinki. All participants provided informed consent and were informed that they could withdraw from the study at any time.

### 2.4. Analyses

All utilities were calculated with UK value sets. Normality of the distributions was assessed with the Kolmogorov–Smirnov test, based on which, comparisons between SF-6D subgroups were made with ANOVA, and between EQ-5D-3L and 5L subgroups with the Kruskal–Wallis test. A minimally important difference (MID) is defined as the smallest difference in score in the domain of interest which patients perceive as beneficial and which would mandate, in the absence of troublesome side effects and excessive cost, a change in the patient’s management [42]. A previous study, based on eleven reviewed studies, determined MIDs of 0.041 and 0.074 for SF-6D and EQ-5D, respectively [43], whereas another found 0.027 and 0.082 [44]. Cohen’s d was chosen as the appropriate effect size measure for mean differences, because it is appropriate if two groups have similar standard deviations and are of the same size. Cohen’s d is defined as the difference between two means divided by their pooled standard deviation, with 0.2–0.5 indicative of a small effect, 0.5–0.8 of a medium effect, and >0.8 of a large effect size [45]. The clinical indexes (DAS-28, DAPSA, and BASDAI) categorized the sample into disease severity subgroups based on disease activity. The agreement was examined with the Intraclass Correlation Coefficient (ICC), based on a two-way mixed model with the absolute agreement, and with ICC’s < 0.40 indicating poor agreement, 0.40–0.75 indicating fair to good agreement, and >0.75 indicating excellent agreement [46]. The Bland–Altman method was also implemented to assess agreement between the utilities. Mean differences between pairs of instruments (i.e., bias) and 95% limits of agreement (mean difference ± 1.96 × SD) were plotted, and it is expected that 95% of differences between the two instruments lie between the levels of agreement [47]. All analyses were performed with IBM SPSS Statistics v.24, and statistical significance was set at ≤0.05.

## 3. Results

Three-hundred and fifteen patients fulfilling the inclusion criteria, out of four hundred who were approached (response rate 78.8%), agreed to participate and completed the survey. The MSK disorder subgroups consisted of 114 patients with RA, 57 with PsA, 49 with AS, and 95 with OsP or OP. Patients’ characteristics for each MSK disorder subgroup are displayed in Table 1.

The mean age for the entire sample was 60.1 ± 12.3 years and the majority (73.0%) was female, mostly due to the high proportion of women in the RA and OsP/OP subgroups. Of the sample, 22.1% had completed a college level of education. Overall, the average duration of the MSK disorder was 7.4 ± 8.1 years. More than half the sample (52.7%) did not have any comorbid conditions, and 19.7% suffered from two or more, mostly hypertension. In terms of disease severity, the majority of patients in the RA, PsA, and AS groups were either in remission or with low disease activity and/or well controlled. In the OsP/OP group, T-scores from BMD tests were between −1.0 and −2.5 for 33 (54.1%) patients, and less than −2.5 for 27 (44.3%) patients, corresponding to OsP and OP, respectively. The disorder subgroups differed by demographic and clinical characteristics, with statistically significant differences and associations observed for all characteristics except the number of comorbidities, thus suggesting heterogeneous subsamples.

Demographics and clinical characteristics were reported separately (Table 2) for the severity subgroups which were formed by the disorder-specific clinical indexes used in this study. It should be noted that the OsP/OP was excluded from this analysis due to the absence (in this study) of a clinical index to categorize the severity of this disorder. The results showed that the severity subgroups were fairly homogenous, as they differed significantly only by gender and BMI classification.

According to the frequency distributions of the utilities, EQ-5D-5L and EQ-5D-3L were left-skewed, as opposed to SF-6D scores, which presented a fairly normal distribution (Figure 1), which was confirmed by the Kolmogorov–Smirnov test (*p* = 0.079). A negligible percentage of respondents (<1.0%) scored the lowest utility with all three instruments, implying the absence of floor effects. However, with both EQ-5D versions, almost 20% of the respondents scored the maximum achievable utility value of one, implying the possible existence of ceiling effects.

The comparison of utilities between subgroups of patients with different MSK disorders showed that patients with AS and OsP/OP had consistently higher utilities with all three instruments, compared to PsA patients, who, in turn, had higher scores than the RA patients. However, none of the differences in utilities between the four patient groups was statistically significant. On the other hand, all the instruments discriminated significantly (*p* < 0.001) and in the expected direction, between adjacent groups of increasing severity, with lower utility scores corresponding to subgroups with higher disease severity (Table 3, Figure 2).

Within the RA, PsA, and AS subgroups, many score differences exceeded the suggested MIDs for SF-6D and EQ-5D, and higher differences were expectedly associated with larger effect sizes. On the other hand, small and insignificant differences between the three utilities were observed within the OsP/OP group. Within disease severity groups, significant differences and large effect sizes were observed for the moderate and high disease activity levels. In these subgroups of reduced health status, SF-6D utilities were higher than EQ-5D-5L, which were higher than EQ-5D-3L.

The ICCs showed that there was excellent agreement between the utilities in the RA, PsA, and AS groups, but only fair to good agreement between SF-6D, and EQ-5D-3L and 5L in the OsP/OP group. For patients with medium or high/active disease status, agreement between EQ-5D-3L and SF-6D, and between EQ-5D-5L and SF-6D was generally low, e.g., ICC = 0.203 between EQ-5D-3L and SF-6D, and ICC = 0.384 between EQ-5D-5L and SF-6D.

On the other hand, the agreement between instruments in patient groups in remission or with low disease activity was excellent. Using the total sample, the points on the Bland–Altman plots are scattered above and below zero (Figure 3), suggesting no consistent bias of one utility instrument versus the other. Mean differences between the utilities differed significantly from zero (on the basis of a one-sample *t*-test), indicating systematic differences (i.e., fixed bias).

In terms of clinical significance, the mean difference between EQ-5D-3L and EQ-5D-5L was 0.038, which is significantly lower (*p* < 0.01) than the suggested MID of 0.074 for EQ-5D. The mean difference between SF-6D and EQ-5D-3L was 0.082, which is comparable to the MID for EQ-5D, but significantly higher (*p* < 0.01) than the suggested MID of 0.041 for SF-6D. The mean difference between SF-6D and EQ-5D-5L was 0.044, which is comparable to the MID of 0.041 for SF-6D, but significantly lower (*p* < 0.05) than the suggested MID for EQ-5D. As shown in Figure 3, the levels of agreement, i.e., the ranges containing 95% of the differences between the two instruments, were multiples of the MIDs, implying clinical significance and that the measurement instruments may not be used interchangeably.

## 4. Discussion

Study samples are almost never uniform in terms of socio-demographic, disease-related, or other characteristics (disease activity or severity), which usually affect the level of health that individual experiences. In an effort to provide more insight into this issue, the present study compared EQ-5D-3L, EQ-5D-5L, and SF-6D utility scores between and within groups of patients with various MSK disorders of different severity, which were determined by valid clinical indexes of disease activity. These differences might have not been directly evident in an overall sample of patients of each MSK disorder. The potential benefit from a “deeper look” into severity subgroups is the emerging evidence for choosing the appropriate utility measure in a particular application and with a particular population group. Previous utility comparison studies have used valid external classification measures to group subjects by severity in disease groups [12,14] and in general population samples [18].

The analyses, in some cases, generated noteworthy utility differences within patient subgroups, many of which were statistically significant and/or exceeded the reported MIDs of the instruments. In utility comparison studies, statistically non-significant differences, which are clinically important, are often ignored. This is particularly common when sample sizes are relatively small. Furthermore, even small differences (i.e., much less than MIDs) in measurements can become statistically significant by increasing the sample, although such small differences could be irrelevant (i.e., no clinical importance) to patients or clinicians [48]. Thus, statistical significance does not necessarily imply clinical importance. For these reasons, MIDs were used in the analyses to enhance the comprehension of the differences between the three utilities.

It should be noted that all utilities in this study were calculated with UK value sets. As our main intention was to simulate disease-specific data to compare the instruments, this methodological choice should not be a significant issue. As in any study, it should be kept in mind when interpreting the results that they are related to the specific value sets which were used to derive them, and that if other country value sets were used, the results would probably be different. Notably, the Greek EQ-5D-3L validation study [32] and a large-scale Greek study reporting SF-6D population norms [18] were also conducted using UK valuations. In general, we are unaware of any Greek study reporting EQ-5D and SF-6D utilities, both in general populations and in disease groups, which used any other (than the UK) value set to derive utilities. In any case, UK valuations are widely used in studies from many countries, and this increases the potential for international comparisons of results.

Our results showed that individuals in remission, i.e., those better off, had similar utility scores with all three instruments. On the other hand, patients with some level of MSK disorder severity had higher mean SF-6D scores, compared to both EQ-5D-3L and EQ-5D-5L, with the absolute values of the differences, as well as their significance, increasing with increasing severity. This could have implications in cost-utility analyses of interventions aiming to treat patients with MSK disorders; for example, in subgroups with higher severity, the SF-6D would most likely generate higher utilities than both EQ-5D-3L and 5L. These discrepancies, which were shown to exceed MIDs, imply significant differences in estimated QALYs. The implication is that one, or perhaps all, the instruments might be generating scores that are not suitable for calculation of QALYs, since they would be providing contradictory estimates of QALY gains from an intervention.

Regarding the comparison between EQ-5D-5L and SF-6D, the results agree with previous studies that compared these two instruments [26,27,28,29]. One recent study [49] compared utilities from EQ-5D-5L and two versions of SF-6Dv2 in patients with breast cancer, and showed that both exceeded EQ-5D-5L, which is also in line with our findings. The comparison between EQ-5D-5L and 3L resulted in the former producing higher utilities in all subgroups. According to the literature, 5L values are expectedly higher because the 3L value set has more states which are considered worse than dead compared to the UK 5L crosswalk value set [50].

Our explanation for the discrepancies between EQ-5D-5L and SF-6D is similar to that given in the literature for the discrepancies between EQ-5D-3L and SF-6D. Firstly, it could be due to the potential ceiling effect of EQ-5D [8]. In this study, 18.2% and 17.4% of the participants reported no problems on all dimensions of EQ-5D-5L and EQ-5D-3L, thus generating a maximum utility of one, whereas the respective figure for SF-6D was only 1.8%. It might also be the different measuring ranges of the two instruments, as SF-6D utilities are limited within the upper and lower bounds of the EQ-5D utilities. Finally, it has been suggested that the different valuation systems used in each instrument might be behind utility score discrepancies [51,52], and that SG generally produces higher values than TTO. This is particularly true in more severe states, whereas TTO is usually higher for milder states [53].

By means of the ICCs, the results showed very good to excellent agreement between utilities in the MSK disorder subgroups. Agreement between SF-6D and the EQ-5D-3L and 5L versions was higher than that reported in a recent study [28]. Our study also showed that agreement between the instruments tends to deteriorate in groups of patients who are worse off, i.e., having higher disease activity levels or lower general health status. This was evident in the medium- and high-disease-activity subgroups, with SF-6D utility scores being consistently higher than EQ-5D-3L and 5L. Lower agreement between EQ-5D-3L and SF-6D in lower health states has also been shown in other studies [12,18].

In this study, the 95% levels of agreement (shown in Bland–Altman plots) were higher than the suggested MIDs for both SF-6D and EQ-5D utilities, which implies that most of the differences between the utilities are clinically significant, and that the particular instruments cannot substitute each other. Therefore, the choice of instrument for measuring health-related utility in studies with patients with MSK disorders can lead to different results in the context of a cost-utility analysis. It should be noted that these types of plots are often tricky with preference-based instruments because scores are bound at one, implying that, by definition, the agreement will be very good for healthy people.

Carryover effects are important to consider, as they can threaten the validity of clinical research results [54]. To avoid this, the participants were administered the SF-6D in-between the two EQ-5D versions to prevent EQ-5D-3L responses from being partly due to the EQ-5D-5L. In terms of EQ-5D, and like in other studies [19,55,56], respondents completed EQ-5D-5L before EQ-5D-3L to avoid the tendency to not choose levels 2 and 4, i.e., the “in-between” options, when the three-level version is completed first. In any case, it is worth considering the potential effect of response bias, where a respondent may react differently to questions based on the order in which they appear in a survey. In a recent study, the differences between utility means were partially attributed to order effect bias [57].

This study has some limitations which should be considered. The cross-sectional design prohibited addressing longitudinal validity, responsiveness and test–retest reliability of the instruments across the severity groups, and the calculation of QALYs gained. Sensitivity was assessed with effect sizes, and future studies should address the longitudinal sensitivity of these instruments. Thirdly, the sample size was limited in some subgroups, and utility comparisons should be considered cautiously.

## 5. Conclusions

In disease groups, severity levels determined by disease-specific indexes can reveal measuring discrepancies which generate utility differences between instruments, and, subsequently, different results in the context of cost-utility analyses. Therefore, utility instruments should not be regarded as interchangeable, and an informed choice on the most appropriate one might require broader within-sample investigations. Our results, which focus on differences in utilities between severity groups, can be seen as a contribution to this objective, and might have implications for economic evaluations of interventions aiming to treat patients with MSK disorders. The discrepancies between utilities, which were shown to mostly exceed MIDs, might imply significant differences in estimated QALYs. In any case, these findings warrant further investigation in future studies involving different disease groups, as well as different health classification systems.

## Figures and Tables

**Figure 1 jcm-11-04097-f001:**
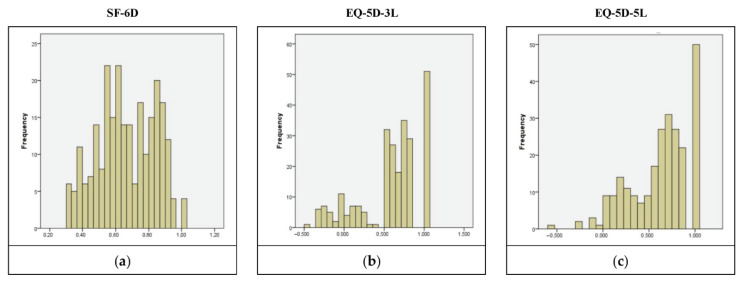
Distribution of SF-6D (**a**), EQ-5D-3L (**b**), and EQ-5D-5L (**c**) utility scores.

**Figure 2 jcm-11-04097-f002:**
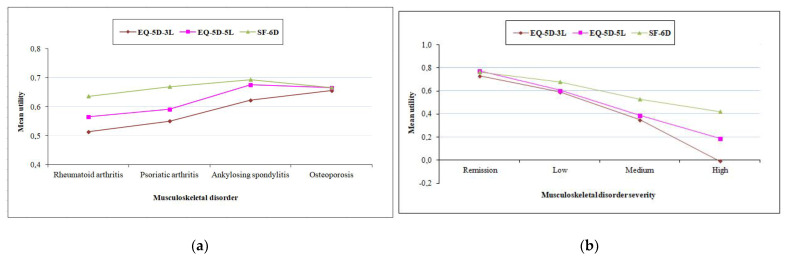
Comparison of utilities between musculoskeletal disorder (**a**) and severity groups (**b**).

**Figure 3 jcm-11-04097-f003:**
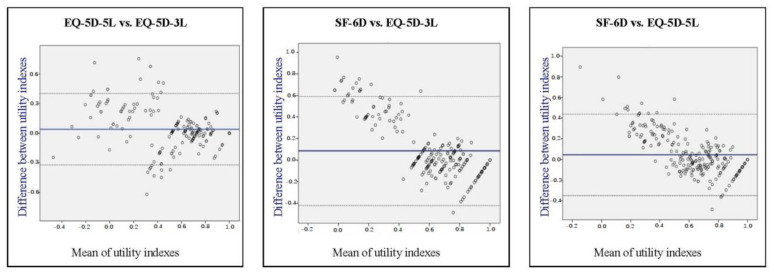
Bland–Altman plots of utility differences between the instruments (in vertical axes).

**Table 1 jcm-11-04097-t001:** Demographic and clinical characteristics of the sample.

Characteristics	Total Sample (N = 315)	Rheumatoid Arthritis (N = 114)	Psoriatic Arthritis (N = 59)	Ankylosing Spondylitis (N = 47)	Osteopenia/ Osteoporosis (N = 95)	*p*-sig.^1^
Female, N (%)	230 (73.0)	94 (82.5)	37 (62.7)	10 (21.3)	89 (93.7)	<0.001
Age, (mean ± SD)	60.1 ± 12.3	62.8 ± 12.5	56.4 ± 12.9	51.0 ± 12.4	63.7 ± 8.4	<0.001
Years disease duration, (mean ± SD)	7.4 ± 8.1	9.2 ± 8.6	7.2 ± 8.2	9.3 ± 9.9	3.9 ± 4.4	<0.001
Educational level, N (% valid)						
Primary	98 (31.9)	52 (46.4)	16 (27.1)	10 (21.3)	20 (22.5)	0.007
Secondary	141 (45.9)	41 (36.6)	29 (49.2)	25 (53.2)	46 (51.7)	
Tertiary	58 (22.1)	19 (17.0)	14 (23.7)	12 (25.5)	23 (25.8)	
BMI Class, N (% valid)						
Normal (18.5–24.9)	126 (41.3)	39 (35.5)	11 (19.3)	21 (45.7)	55 (59.8)	<0.001
Overweight (25.0–29.9)	114 (37.4)	42 (38.2)	20 (35.1)	19 (41.3)	33 (35.9)	
Obese (≥30.0)	65 (21.3)	29 (26.4)	26 (45.6)	6 (13.0)	4 (4.3)	
Comorbidities, N (%)						
None	166 (52.7)	52 (45.6)	30 (50.8)	34 (72.3)	50 (52.6)	0.103
One	87 (27.6)	34 (29.8)	18 (30.5)	9 (19.1)	26 (27.4)	
Two or more	62 (19.7)	28 (24.5)	11 (18.7)	4 (8.5)	19 (20.0)	
Disorder severity, N (% valid)						
Remission/Controlled	113 (54.3)	49 (45.4)	28 (50.0)	36 (81.8)	-	
Low	31 (14.9)	16 (14.8)	15 (26.8)	-	-	<0.001
Moderate	33 (15.9)	23 (21.3)	10 (17.9)	-	-	
High/Uncontrolled	31 (14.9)	20 (18.5)	3 (5.4)	8 (18.2)	-	

^1^ According to ANOVA for age and disease duration, and chi-squared test for gender, education, BMI, comorbidities, and severity.

**Table 2 jcm-11-04097-t002:** Demographic and clinical characteristics of the sample by disease severity ^1^.

Characteristics	Remission (N = 113)	Low Severity (N = 31)	Moderate Severity (N = 33)	High Severity (N = 31)	*p*-sig. ^2^
Female, N (%)	61 (54.0)	28 (90.3)	24 (72.7)	20 (64.5)	0.002
Age, (mean ± SD)	58.1 ± 14.1	58.2 ± 11.3	61.3 ± 12.1	58.6 ± 11.1	0.701
Years disease duration, (mean ± SD)	8.8 ± 8.5	9.9 ± 8.8	8.6 ± 8.6	7.0 ± 10.3	0.668
Educational level, N (% valid)					
Primary	34 (30.6)	15 (48.4)	15 (45.5)	11 (35.5)	0.437
Secondary	50 (45.0)	10 (32.3)	14 (42.4)	14 (45.2)	
Tertiary	27 (24.3)	6 (19.4)	4 (12.1)	6 (19.4)	
BMI Class, N (% valid)					
Normal (18.5–24.9)	48 (44.4)	5 (16.1)	7 (22.6)	9 (29.0)	0.017
Overweight (25.0–29.9)	33 (30.6)	17 (54.8)	11 (35.5)	15 (48.4)	
Obese (≥30.0)	27 (25.0)	9 (29.0)	13 (41.9)	7 (22.6)	
Comorbidities, N (%)					
None	64 (56.6)	12 (38.7)	14 (42.4)	21 (67.7)	0.080
One	33 (29.2)	9 (29.0)	10 (30.3)	4 (12.9)	
Two or more	16 (14.2)	10 (32.3)	9 (27.3)	6 (19.4)	

^1^ Determined by DAS-28 for RA, DAPSA for PsA, and BASDAI for AS. OsP/OP sample excluded from the analysis. ^2^ According to ANOVA for age and disease duration, and chi-squared test for gender, education, BMI, and comorbidities.

**Table 3 jcm-11-04097-t003:** Comparison of utility scores between and within MSK disorder and severity groups.

MSK Disorder	Between Groups Comparisons ^a^								Within Groups Pairwise Comparisons/Mean Difference ^b^ (Effect Size) [ICC]			
	N	EQ-5D-3L	N	EQ-5D-5L	N	SF-6D	N	EQ-5L/EQ-3L	N	SF-6D/EQ-3L	N	SF-6D/EQ-5L
Rheumatoid arthritis	105	0.513	112	0.565	101	0.635	103	0.057 ** (0.161) [0.926 ***]	96	0.125 *** (0.400) [0.738 ***]	101	0.073 *** (0.284) [0.834 ***]
Psoriatic arthritis	57	0.550	55	0.591	53	0.668	54	0.027 (0.069) [0.954 ***]	52	0.109 ** (0.348) [0.773 ***]	51	0.082 * (0.267) [0.832 ***]
Ankylosing spondylitis	41	0.622	45	0.675	44	0.693	41	0.064 * (0.199) [0.923 ***]	41	0.078 * (0.266) [0.804 ***]	43	0.021 (0.087) [0.868 ***]
Osteopenia/Osteoporosis	82	0.655	87	0.665	75	0.666	78	0.011 (0.039) [0.861 ***]	70	0.021 (0.092) [0.635 ***]	70	−0.003 (0.014) [0.674 ***]
		*p* = 0.061		*p* = 0.071		*p* = 0.327						
Severity group ^c^												
Remission/Inactive	105	0.731	107	0.753	100	0.750	102	0.020 (0.078) [0.951 ***]	96	0.010 (0.038) [0.823 ***]	98	−0.005 (0.029) [0.856 ***]
Low	28	0.504	30	0.628	27	0.683	27	0.003 (0.012) [0.857 ***]	25	0.062 (0.308) [0.811 ***]	26	0.058 (0.285) [0.858 ***]
Moderate	30	0.349	32	0.409	29	0.544	29	0.053 (0.161) [0.912 ***]	28	0.192 * (0.695) [0.511 *]	29	0.131 * (0.590) [0.591 **]
High/Active	28	−0.006	31	0.186	30	0.423	28	0.197 *** (0.657) [0.704 ***]	28	0.429 *** (1.820) [0.203]	30	0.249 *** (1.214) [0.384 *]
		*p* < 0.001		*p* < 0.001		*p* < 0.001						

Note: N corresponds to the sample size for which the utilities were derived (non-missing data) or compared pairwise. ^a^ Significance tested with ANOVA for SF-6D and Kruskal–Wallis test for EQ-5D-3L and 5L. ^b^ Significance tested with Wilcoxon test. ^c^ According to DAS28 for RA, DAPSA for PsA, and BASDAI for AS. * *p* < 0.05, ** *p* < 0.01, *** *p* < 0.001.

## Data Availability

Not applicable.

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
