# Peer review of "A Comparison of EQ-5D-3L, EQ-5D-5L, and SF-6D Utilities of Patients with Musculoskeletal Disorders of Different Severity: A Health-Related Quality of Life Approach"

_jcm, 2022, doi:10.3390/jcm11144097_

Round 1
Reviewer 1 Report
Thank you to the authors for the privilege of reviewing their work. I have been upfront with editors that I would not consider myself an expert in survey/measure validation or comparison. With that said, I am familiar with the 3 measures the authors compared and broadly familiar with the analytic techniques applied in this study. I found the manuscript well-written and fairly easy to understand (there are several minor typos in the Results and Discussion that required 2-3 readings to make sure I was understanding correctly). I found the background about the 3 utility measures to be quite succinct and useful. Similarly, the methods were very well-written and I had to methodologic concerns.
I think my main concern was the justification of the manuscript in the introduction and inferences of application (based on the authors' findings) in the discussion. In fact, I'm still a bit unclear on why the authors chose to compare measure agreement by severity ACROSS several diseases. I would understand the rationale if authors chose to examine agreement by severity within a single MSK disorder (remission vs low vs moderate-severe RA), but I think the decision to combine RA with psoriatic arthritis and AS participants was questionable. I think authors need to explicitly justify this decision and explain why they don't believe this makes interpretation of their findings more difficult.
Similarly, for the discussion, there was a lot of "jargony" discussion related to survey methodology and analytic techniques. that belongs in the methods and made discussion hard to follow. I also found a good portion of the discussion was devoted to restating of their results and whether this was consistent with existing literature. This aspect of discussion can be shortened and simplified. Instead, I wanted to see more of the discussion focused on what authors believe are clinical/research IMPLICATIONS of their findings. For example, describe scenarioes in which authors would recommend choosing different utility measures.
Some minor comments:
-Do authors have any data on # eligible participants approached, % enrolled/consented, and % of consented who completed survey. This has important bearing of validity of findings.
-Can a caption be added to Figure 3 Bland Altman plot so that non-survey methodologists could interpret?
- Page 7 Line 236, is this mislabled Table 2? Should be Table 3...
Reviewer 2 Report
Dear Authors,
the paper is well presented and the topic is well studied.
Only details of the ethical committee should be reported in the text.
Author Response
Thank you for reviewing our manuscript.
A subsection titled "2.3. Ethical issues" has been added (page 4) which details all issues related to ethics.
Reviewer 3 Report
A comparison of EQ-5D-3L, EQ-5D-5L and SF-6D utilities of patients with musculoskeletal disorders of different severity: An approach to Health quality of life
Quality of life is an important issue in every aspect of patients’ treatment. A good approach with the proper survey gives us a reliable outcome. As shown by the authors the instruments to measure the HQoL are not interchangeable, and we should properly use them when we plan a study.
I find this study interesting well organized, statistically well prepared. The described patient groups are well defined. The studied volume of the patents is satisfactory.
I have only 1 question:
1. What was the response rate?
It is recommended to continue the study and a present larger group of cases and to compare the outcomes between the different clinical centers and populations. It might inform us how to find or prepare “the perfect” survey.
I recommend this manuscript be published as an original article.
Author Response
Thank you for reviewing our manuscript. The response rate has been added in the first paragraph of the Results section. We agree with your suggestion to continue the study, and hope to address larger and more diverse patient samples in the future.